# A Heme-Binding Transcription Factor BACH1 Regulates Lactate Catabolism Suggesting a Combined Therapy for Triple-Negative Breast Cancer

**DOI:** 10.3390/cells11071177

**Published:** 2022-03-31

**Authors:** Joselyn Padilla, Bok-Soon Lee, Karen Zhai, Bethany Rentz, Tia Bobo, Norca Maritza Dowling, Jiyoung Lee

**Affiliations:** 1Department of Biochemistry and Molecular Medicine, School of Medicine and Health Sciences, Washington, DC 20037, USA; joselynp@gwu.edu (J.P.); blee2021@gwu.edu (B.-S.L.); karenzhai0331@gmail.com (K.Z.); 2Animal Research, Office of the Vice Provost for Research, 1922 F Street NW, Washington, DC 20052, USA; brentz@gwu.edu (B.R.); tbobo@gwu.edu (T.B.); 3Department of Acute & Chronic Care, School of Nursing, Washington, DC 20037, USA; nmdowling@gwu.edu; 4Department of Epidemiology, Milken Institute School of Public Health, Washington, DC 20052, USA; 5Center for Aging, Health and Humanities, Washington, DC 20006, USA; 6GW Cancer Center, The George Washington University, Washington, DC 20037, USA

**Keywords:** lactate catabolism, BACH1, lactate transporter, novel combination therapy targeting BACH1 and MCT1, triple-negative breast cancer

## Abstract

The oncogenic expression or mutation of tumor suppressors drives metabolic alteration, causing cancer cells to utilize diverse nutrients. Lactate is a known substrate for cancer cells, yet the regulatory mechanisms of lactate catabolism are limited. Here, we show that a heme-binding transcription factor, BACH1, negatively regulates lactate catabolic pathways in triple-negative breast cancer (TNBC) cells. BACH1 suppresses the transcriptional expression of monocarboxylate transporter 1 (MCT1) and lactate dehydrogenase B, inhibiting lactate-mediated mitochondrial metabolism. In our studies, the depletion of BACH1 either genetically or pharmacologically increased the lactate use of TNBC cells, increasing their sensitivity to MCT1 inhibition. Thus, small inhibitory molecules (SR13800 and AZD3965) blocking MCT1 better suppressed the growth of BACH1-depleted TNBC cells than did the controls. Particularly, hemin treatment degrading BACH1 proteins induced lactate catabolism in TNBC cells, generating synthetic lethality with MCT1 inhibition. Our data indicates that targeting BACH1 generates metabolic vulnerability and increases sensitivity to lactate transporter inhibition, suggesting a potential novel combination therapy for cancer patients with TNBC.

## 1. Introduction

Aerobic glycolysis has long been a hallmark of cancer metabolism, because cancer cells increase glucose uptake and lactate production in the presence of oxygen [1,2]. The increased glucose uptake of cancer cells occurs through oncogenic mutation to support the rapid growth and proliferation of cancer cells [3,4]. With limited glucose and oxygen in the tumor microenvironment, cancer cells use alternative energy substrates to meet their bioenergetic demands or balance oxidative stress [5,6]. During these metabolic adaptations, cancer cells depend on oncogenic signaling or environmental cues, displaying flexible metabolism for cell survival [7,8]. The metabolic flexibility generates the resiliency of cancer cells, while it simultaneously contributes to the resistance or inefficacy of drugs targeting metabolic pathways, including the serine biosynthesis or guanosine biosynthesis pathways of cancer cells [8]. Therefore, understanding cancer metabolism and its regulatory mechanism is extremely important for obtaining maximum efficacy of metabolism-targeting cancer drugs.

Lactate derived from tumors after aerobic glycolysis promotes tumorigenesis, including tumor invasion and metastasis, and predicts patients’ outcomes [5,9]. Anabolic pathways of lactate are upregulated by oncogenes such as hypoxia-inducing factor 1 alpha (HIF1α) and Myc in cancer cells [10,11]. Although lactate had previously been considered as cellular waste, it is actually a proven energy substrate for both aerobic cancer cells and immune cells in glucose-deprived conditions [12,13,14,15]. In lung tumors, flux tracing analysis showed that cancer cells favored the use of lactate over glucose [15]. Moreover, lactate-utilizing tumor cells enhanced antioxidant metabolic pathways to support cell survival during the circulation of cancer cells, and ultimately, enabled efficient metastasis [9].

As an initial step for lactate catabolic pathways using extracellular lactate, cells utilize a family of monocarboxylate transporters (MCTs), encoded by *SLC16A* genes [15,16,17]. MCT1 expression is induced in malignant tissues and associated with a worse prognosis of cancer patients, including those with breast cancer [18,19,20,21]. Mechanistically, MCT1 transcription is increased due to the loss of p53 and NF-kB, as p53 directly interacts with the promoters of MCT1 genes for transactivation under hypoxia [22]. In addition, MCT1 co-expresses with CD147 on the cell surface or is directly activated by oncogenic Myc by binding at E-boxes [19,23,24]. Intracellular lactate is then either incorporated into mitochondria through the mitochondrial transporters, MCT2 or MCT3, or converted into pyruvate by lactate dehydrogenase B (LDHB). Further, pyruvate is transported by mitochondrial pyruvate carriers (MPCs) for the tricarboxylic acid cycle to fuel mitochondrial respiration [25,26]. The transporters that mediate either the efflux or influx of lactate has become a useful therapeutic target to treat cancer patients [13,14]. Preclinical studies identified that the blockade of MCT1 resulted in reduced metastasis or primary tumor growth, showing anti-cancer activity in multiple cancer models [13,18,27,28]. A small MCT1 inhibitory molecule, AZD3965, is currently in early-stage clinical trials for cancer treatment [29], yet little is known about how lactate catabolic pathways are regulated in cancer cells.

A heme-regulating transcription factor, BTB and CNC homology 1 (BACH1), is more highly expressed in triple-negative breast cancer (TNBC), the most aggressive subtype of breast cancer [30,31]. BACH1 is stabilized by antioxidants or in lung adenocarcinoma that is co-mutated in KRas, p53, and Keap1 [32,33]. BACH1 promotes the invasion and metastasis of cancer cells and is also associated with the poor prognosis of cancer patients with TNBC or lung tumors. In recent reports, BACH1 showed that it suppresses mitochondrial oxidative phosphorylation, but enhances aerobic glycolysis in TNBC or lung adenocarcinoma [32,33,34]. At the transcriptional levels, BACH1 directly regulates the expression of the mitochondrial electron transport chain (ETC) genes pyruvate dehydrogenase kinases, hexokinase 2 and glyceraldehyde 3-phosphate dehydrogenase in cancer cells. Since BACH1 is a heme-binding transcription factor for cellular heme homeostasis, heme (hemin) was used as a tool to reduce BACH1 protein levels in numerous tumor models [32,33,34]. Hemin treatment induced the ubiquitin-dependent degradation of BACH1 protein, reprogrammed cancer metabolism, and generated cancer vulnerability against mitochondrial oxidative phosphorylation inhibitors. Importantly, hemin is a non-toxic ingredient of Panhematin, an FDA-approved drug for the treatment of acute porphyria, suggesting that hemin could be clinically useful for patients with cancers [35,36,37].

In this study, we assessed how breast cancer cells utilize lactate as a substrate in a microenvironment with limited glucose. We identified that BACH1 is an endogenous regulator of lactate metabolic pathways in TNBC cells. Combined analyses of transcriptomic profiling, gene expression analyses, and metabolic phenotypes using BACH1-depleted TNBC cells indicated that BACH1 suppresses MCT1 and LDHB expression and the subsequent lactate utilization of TNBC cells. Therefore, depleting BACH1 using either shRNA or a non-toxic inhibitor hemin could elicit the lactate dependency of cancer cells, increase the efficacy of MCT1 inhibitors, and suggest better therapeutic options for those with TNBC.

## 2. Materials and Methods

### 2.1. Cell Lines

Breast cancer cell lines (MDA-MB-231, MDA-MB-436, HCCC1937, and 4T1.2) were obtained from ATCC and cultured according to the manufacturer’s instructions. BM1 cells (MDA-MB-231-Bone Metastatic subpopulation) were obtained from Dr. Rosner’s lab. [34]. MDA-MB-231, MDA-MB-436, BM1, and 4T1.2 cells were maintained in high glucose DMEM (40 g glucose, 2 mM glutamine, Gibco, Waltham, MA, USA, #11965118) supplemented with 10% heat-inactivated Fetal Bovine Serum (FBS, Gibco, #10437028) or dialyzed FBS (GeminiBio, West Sacramento, CA, USA, #100–108) and 50 U penicillin/streptomycin solution (Gibco #15140428). HCC1937 cells were maintained in RPMI 1640 (Gibco, #11875085) supplemented with 10% heat-inactivated FBS and 50 U penicillin/streptomycin solution. For in vitro assays, the culture medium was changed to glucose-limited DMEM (1.25 mM glucose, 2 mM glutamine, 10% FBS) with 4 mM Sodium L-lactate (Sigma Aldrich, St. Louis MO, USA, #1614308). The stable knock-down of BACH1 was performed using a lentiviral particles-carrying shRNA targeting BACH1 (Santa Cruz, Dallas TX, USA sc-37064-v) or a scrambled shRNA control (Santa Cruz, sc-108080), and further selected using puromycin (10 μg/mL) for 10 days, as previously described [34]. The protein levels of BACH1 were validated using Western blotting before further experimentation. For rescued expression of Bach1 (WT-Bach1 cell lines), the shBACH1 TNBC cells were transduced using lentiviral particles carrying murine Bach1 and further selected using hygromycin (10 μg/mL) for 10 days. For transient expression of BACH1 in shBACH1 cells, BACH1-strep plasmid (pcDNA3.1-2xFLAG-2xSTREP-BACH1, Addgene, Watertown, MA, USA) was transduced in BACH1-depleted cells using Lipofectamine 3000 Transfection Reagent (Invitrogen, Waltham, MA, USA, L300001) in OPTI-MEM (Gibco, #11058021). All breast cancer cells were routinely monitored for mycoplasma using a MycoAlert Mycoplasma detection kit (Lonza, Basel, Switzerland, #LT07-218) and authenticated by short tandem repeat analysis.

### 2.2. Lactate Dehydrogenase B Activity Assays

Lactate dehydrogenase B activity was monitored using a lactate dehydrogenase B activity assay kit (Abcam, Cambridge, United Kingdom, ab140361). Simply, harvested cancer cells in the extraction buffer were used for assays and quantified according to the manufacturer’s instructions.

### 2.3. Lactate Assays

BACH1-depleted or hemin-treated TNBC cells were cultured in phenol-red free DMEM (Gibco, #1443001) containing 10% dFBS and lactate (5 mM). Media collected before and after the culture of the cells for 18 h were used for the quantification of lactate using an L-lactate assay kit (Abcam, ab65331) according to the manufacturer’s protocol.

### 2.4. qRT-PCR

The total RNA was isolated from the cells using TRIzol reagent (Invitrogen, #15596026) and quantified using a NanoDrop one/One^C^ Microvolume UV/Vis spectrophotometer (ThermoFisher, ND-ONE-W). A t total of 2 μg of RNA was adapted for the reverse transcriptase reaction using a High Capacity cDNA Reverse Transcription Kit (ThermoFisher, #4368814) to generate the cDNA library. Real-time PCR was carried out using a Fast Start Essential DNA Master mix (2X) reagent and primers using a CFX96 Touch Real-Time PCR detection system (BioRad, GWU). Cq values normalized relative to the expression of endogenous control genes (36B4) using 2 ^(−ΔΔCq)^ were plotted. The primer pairs used are shown as follows:

BACH1-Forward: CAC CGA AGG AGA CAG TGA ATC CBACH1-Reverse: GCT GTT CTG GAG TAA GCT TGT GCCD147-Forward: GGC TGT GAA GTC AGA ACA CCD147-Reverse: ACC TGC TCT CGG AGC CGT TCASLC16A1 (MCT1)-Forward: AGG TCC AGT TGG ATA CAC CCCSLC16A1 (MCT1)-Reverse: GCA TAA GAG AAG CCG ATG GAA ATSLC16A3 (MCT4)-Forward: CCA TGC TCT ACG GGA CAG GSLC16A3 (MCT4)-Reverse: GCT TGC TGA AGT AGC GGT TMPC1-Forward: TTA TCA GTG GGC GGA TGA CATMPC1-Reverse: GCT GTA CCT TGT AGG CAA ATC TCMPC2-Forward: CCT CCA GCC CGA GGG ACC TTTMPC2-Reverse: CAT CGC CGA GGG ATC GLDHA-Forward: ATG GCA ACT CTA AAG GAT CAG CLDHA-Reverse: CCA ACC CCA ACA ACT GTA ATC TLDHB-Forward1: CCT CAG ATC GTC AAG TAC AGT CCLDHB-Reverse1: ATCACGCGGTGTTTGGGTAATLDHB-Forward2: TCC CGT GTC AAC AAT GGT AALDHB-Reverse2: CCC ACA GGG TAT CTG CAC TT36B4-Forward: GGA CAT GTT GCT GGC CAA TAA36B4-Reverse: GGG CCC GAG ACC AGT GTT

### 2.5. Immunoblotting

Whole-cell or tumor lysates were prepared using RIPA buffer (Sigma Aldrich, St. Louis, MO, USA, R2078) with protease inhibitor cocktail set III (Millipore, Burlington MA, USA, 539134) at 4C and quantified using a Pierce^TM^ BCA Protein Assy Kit (ThermoFisher, #23225) before blotting. Lysate samples were loaded onto 7.5% SDS acrylamide gels using a TGX Stain-free FastCast Acrylamide kit 7.5% (BioRad, Hercules, CA, USA, #1610181) for sample separation. Protein gels were transferred to the precut nitrocellulose membrane (Biorad, #1620150) and blotted using the following primary and secondary antibodies (dilution): BACH1 (Santa Cruz, Dallas, TX, USA. sc271211, 1:500), MCT1 (Millipore AB3538P, 1:1000), Alpha-Tubulin (ThermoFisher A11126, 1:5000), Beta-Actin (Sigma Aldrich Sab5600204, 1:10,000), LDHB (Abcam, Cambridge, United Kingdom ab53292, 1:1000), MPC2 (Millipore MABS1914, 1:1000 and Sigma Aldrich HPA056091, 1:1000), HRP-conjugated antibody against mouse (Santa Cruz sc2005, 1:5000), HRP-conjugated antibody against rabbit (Santa Cruz sc2357, 1:5000), or IRDye conjugated secondary antibodies (Licor, Lincoln, ME, USA, 1:5000) against mouse or rabbit. Blots were imaged using ChemiDoc MP (BioRad, GWU) and quantified using Image J software.

### 2.6. Chemical Reagents

AZD3965 (Selleckchem, Houston, TX, USA, S7339, APexBio, Houston, TX, USA, C5049), SR13800 (Tocris, Bristol, United Kingdom, #5431), GSK2837808A (Millipore Sigma 5336600001), Hemin (Sigma Aldrich 51280), and UK5099 (Selleckchem S5317) were used as indicated. For cell line experiments, AZD3965 (100–200 μM in DMSO), SR13800 (25–50 μM in DMSO), GSK2837808A (100–200 μM in DMSO), UK5099 (100 μM in DMSO), and hemin (20 μM in 20 mM of NaOH, filter sterilized) were used as a final concentration. DMSO less than 5% of total volume or NaOH (20 mM, sterilized using 0.45 μm filters) were used as vehicle controls.

### 2.7. Viability Assays

Breast cancer cells (8–10 × 10^3^ per well) were plated on 96-well plates, as previously described [34]. After 24 h of plating, inhibitors (as indicated) were added in the culture media (2 mM glutamine, 4 mM lactate), containing either low (1.25 mM) or high (25 mM) glucose, for 48–72 h. Viable cells were stained using 2 nM of Calcein AM dye (R&D systems, #5119) in 1x phosphate buffered saline (PBS) for 1 h at 37 °C to measure absorbance with excitation at 420 nm and emission at 520 nm using a Fisher Victor3 Spectrophotometer (Genomic Core, GWU). The absorbance was used to reflect live-cell numbers and was normalized to those in the control or with vehicles. The results are shown as relative viability (%). For colony formation assays, breast cancer cells (3–5 × 10^3^ cells per well) were plated on 24-well plates and treated with inhibitors as indicated in the culture media (2 mM glutamine, 4 mM lactate), containing either low (1.25 mM) or high (25 mM) glucose. After 48–72 h of incubation, the media was re-filled with the regular culture media (25 mM glucose, 2 mM glutamine, 4 mM lactate) and incubated for another 10 days. Colonies were stained using crystal violet solution (0.01% *w*/*v*) for 5 min and washed with dH_2_O twice before imaging. Stained cells (in triplicate) were quantified using ImageJ software.

### 2.8. siRNA

LDHB siRNA (Millipore Sigma, MISSION esiRNA EHU108541) and scrambled siRNA were transfected into the cells using Lipofectamine 3000 reagent (Thermo Fisher L3000015) in OPTI-MEM (Invitrogen) overnight for in vitro assays.

### 2.9. Seahorse Analysis

Mitochondrial metabolic phenotypes were determined using a Seahorse Bioscience Analyzer XFe96 Analyzer (Agilent, Santa Clara CA, USA, GWU), according to the manufacturer’s instructions. Hemin-pretreated breast cancer cells or shBACH1 cells (5 × 10^4^ cells per well) were plated in 96-well plates for at least 18 h. On the following day, the culture medium was changed to a base medium (DMEM, 143 mM NaCl, phenol red, pH 7.35) and incubated for 1 h at 37 °C without CO_2_. Cells were injected with lactate (4 mM) or pyruvate (3 mM) to monitor the oxygen consumption rate (OCR) and the extracellular acidification rate (ECAR) every 3 min. BCA protein assays were used to normalize the metabolic rates to the cell number.

### 2.10. Chromatin Immunoprecipitation Assays

Breast cancer cells (8 × 10^6^ cells) in 15 cm plates were crosslinked with 10% formaldehyde for 10 min followed by quenching with glycine (0.125 mM) for 3 min. After washing the cells with cold PBS, the total cell lysates were placed on ice, sonicated (50% input for 10 s with a 10 s pause for 4 cycles) for DNA shredding, and further precipitated using antibodies (2 μg/cell lines) against BACH1 (R&D Systems; AF5776, Santa Cruz, sc271211), RNA polymerase II phosphoS5 (Abcam, ab5131), Histone lysine 27 trimethylation (Abcam, ab6002), and normal IgG (Santa Cruz, sc-2028 and sc-2025) overnight at 4 °C. The chromatids pulled-down using protein A/G PLUS-agarose (Santa Cruz, sc-2003) were washed for further DNA isolation and qPCR using primers, as previously performed.

ChIP PCR primers:

MCT1 Region I-forward: CCTCGTTTGCTTGTTCCAGTMCT1 Region I-reverse: CAGTTCGGATGTCTGTGTGGMCT1 Region II-forward: GCCAGTGAACAAAGTGCTGAMCT1 Region II-reverse: CATGCAAATTCGAGAATGGALDHB Region I-forward: ATTGAGGCTCAGGGGAACTTLDHB Region I-reverse: ACTTTGGAGTCAGGCAGCCLDHB Region II-forward: GGCAAGTGTCTGTTAAAGGTGTLDHB Region II-reverse: TCTCTTAAAACCTCTGCAGCAG

### 2.11. Animal Experiments

Animals were maintained in an AAALAC-accredited facility in accordance with the Guide for the Care and Use of Laboratory Animals. All procedures for animal use were approved by the GWU Institutional Animal Care and Use Committee. Athymic nude mice (5-week-old females, Jackson Laboratory and Charles River) were orthotopically injected with human breast cancer cells (BM1, BM1-shBACH1, or shControl, 2 × 10^6^ cells per mouse) in the mammary fat pads. When tumors were palpable, mice were allocated to the random treatment groups. Mice that failed to form tumors were excluded from the experiments. Drug treatment with AZD3965 was blinded, but tumor measurement was not blinded. AZD3965 (30 mg/kg body weight in 0.5% [hydroxypropyl] methyl cellulose, 0.2% Tween80, and 5% DMSO) was provided by oral gavage daily for 17 days. Drug treatment of hemin and SR13800 was not blinded. Hemin (50 mg/kg body weight/day) or vehicle (50% PBS, 50% NaOH 20 mM) were first administered by intraperitoneal injection daily for 7–10 days and followed by SR13800 (30 mg/kg body weight/day) by intraperitoneal injection for the indicated periods. Tumor volumes were measured weekly using a caliper, and body weights were monitored bi-weekly. Tumor volume was calculated using an equation = length × (diameter)^2^ × 0.4.

### 2.12. Lung Metastasis

Lung tissues isolated from the xenografted mice with treatment were fixed using 4% formaldehyde and evaluated by serial sectioning every 100 µm for hematoxylin and eosin (H&E) staining (Vitro Vivo Biotech, Rockville, MD, USA) for the visualization of lung metastases under a microscope (Evos M5000 cell imaging system, ThermoFisher).

### 2.13. Intravasation Assay

For the quantification of the intravasation of cancer cell, whole bloods isolated from the xenografted mice were incubated in 1× Red Blood Cell Lysis buffer (Invitrogen, #00-4333-57) for 30 min at room temperature. The collected circulating cells after centrifugation at 300× *g* at 4 °C were used for RNA isolation and further qPCR using primers to quantify human cancer cells and mouse cells. Primers were purchased from OriGene (Rockville, MD, USA).

Human GAPDH-Forward: GTC TCC TCT GAC TTC AAC AGC GHuman GAPDH-Reverse: ACC CTG TTG CTG TAG CCA AMouse Gapdh-Forward: CAT CAC TGC CAC CCA GAA GAC TGMouse Gapdh-Reverse: ATG CCA GTG AGC TTC CCG TTC AG

### 2.14. Statistical Analyses

Viability assays, ChIP assays, qRT-PCR, and tumor sizes were analyzed to compare the values measured in control groups relative to shBACH1 or hemin treatment by two-tailed Student’s *t*-test using GraphPad Prism software. In vitro experiments were independently repeated at least three times. Data are shown as mean ± s.e.m., unless otherwise noted. A paired Student’s *t*-test or two-way ANOVA was used for multiple treatment comparisons of animal experiments. No statistical approach was used for sample-size determination. Investigators were blinded for drug treatment (AZD3965) by oral gavage of animals, but not blinded for animal allocation and tumor assessment. All in vitro experiments were shown as representative of three or more biological replications, and all animal experiments were performed one time.

## 3. Results

### 3.1. BACH1 Suppresses Lactate Uptake and MCT1 Expression

A highly enriched transcriptional factor BACH1 is a metabolic regulator in TNBC [34]. We assessed whether BACH1 controls substrate utilization for TNBC metabolism. Initially, we screened BACH1 expression levels between selected TNBC cell lines (Appendix A). Amongst TNBC cell lines we tested, MDA-MB-231-BM1 (BM1) cells expressed the highest levels of BACH1 [38,39]. Two TNBC cell lines that express high levels of BACH1, MDA-MB-436 (MB436) and MDA-MB-231 (MB231), were also selected for this study [34]. BACH1 was stably knocked-down using shRNA-carrying lentiviral particles in TNBC cells. To determine whether BACH1 changes carbon substrates other than glucose, we first explored the transcriptomic profiles of BACH1-depleted BM1 cells. Stable BM1-shBACH1 and control shRNA (shCont) cells were cultured in growth media containing lactate (4 mM), low glucose (1.25 mM), and glutamine (2 mM), and their RNA was sequenced (Novogene). *SLC16A1*, which encodes monocarboxylate transporter 1 (MCT1), was significantly, but slightly, increased in the shBACH1 cells as compared to controls (Appendix A). As lactate is the most abundant carbon source in the tumor microenvironment, lactate consumption was monitored using stable TNBC cell lines depleted of BACH1. BACH1-depleted cells (BM1-shBACH1 and MB436-shBACH1) consumed more lactate provided in the growth media (DMEM 1.25 mM glucose, 2 mM glutamine, 5 mM lactate) than control cells (Figure 1A). Subsequently, the BACH1-depleted stable cell lines (BM1, MB231, and murine TNBC 4T1.2 cells) indicated increased levels of MCT1 protein than their controls (Figure 1B). Additionally, MCT1 mRNA levels were increased after BACH1 depletion using siRNA in MB231 and human TNBC HCC1937 cells (Appendix A). Since intracellular lactate is further utilized by lactate dehydrogenase B (LDHB) to facilitate mitochondrial metabolism (Figure 1C), we then measured the LDHB expression in TNBC cells depleted of BACH1. These cells also showed increased LDHB protein expression when compared to control cells (Figure 1B).

As a transcription factor, BACH1 likely regulates the expression of the target genes including, *SLC16A1*, by binding to the promoter areas. In silico motif screening and information from the ENCODE database predicted potential BACH1 binding both upstream (−685) and downstream (+5425) from the transcription start site (TSS) of the *SLC16A1* gene (Figure 1D). Chromatin immunoprecipitation (ChIP) assays using TNBC cells show that BACH1 incorporation is markedly enriched in both the upstream and downstream binding regions of *SLC16A1*, co-enriched with a transcription-suppression histone marker, histone lysine 27 trimethylation (H3K27me3) (Figure 1E and Appendix A). Moreover, BACH1 binding to the upstream (−1066) from the TSS of *LDHB* was significantly enriched compared with control IgG in BM1 and MB436 cells, whereas BACH1 did not interact with the downstream (+4350) of the TSS of *LDHB* (Figure 1D,F). Consistent with these results, BACH1-depleted BM1 and MB436 cells showed a higher activity of LDHB that converts lactate into pyruvate in the lactate catabolic pathways than in the control cells (Figure 1G and Appendix A). These data indicate that BACH1 suppresses expression of SLC16A1 and LDHB at the transcriptional level in TNBC cells.

Furthermore, we determined whether lactate contributes to mitochondrial respiration of TNBC cells by monitoring the mitochondrial oxygen consumption rate (OCR). The BM1-shBACH1 and MB436-shBACH1 cells displayed increased basal OCR, but decreased extracellular acidification rate (ECAR), an indicator of aerobic glycolysis, compared with their shCont cells (Figure 1H and Appendix A). When lactate (4 mM) was infused as a sole carbon source, the BM1-shBACH1 cells showed increased OCR, although not after the addition of pyruvate or glutamine (Appendix A). Lactate addition also decreased the ECAR of BM1 and MB436-shBACH1 cells (Appendix A). However, the expression of the mitochondrial pyruvate carriers 1 and 2 (MPC1 and MPC2) that transport pyruvate into mitochondria for lactate oxidation was not consistently altered by BACH1 silencing in TNBC cells (Appendix A). Collectively, these results suggest that BACH1 suppresses lactate use for mitochondrial respiration through inhibiting MCT1 and LDHB expression in TNBC cells (Appendix A).

### 3.2. BACH1 Depletion Increases Sensitivity to MCT1 Inhibitors for TNBC Viability

Since BACH1-depleted TNBC cells use more lactate for mitochondrial respiration through MCT1 induction than the control cells, we hypothesized that the blockade of lactate transporter MCT1 would be critically lethal to BACH1-depleted cells. To test this hypothesis, we used the small inhibitory molecules SR13800 and AZD3965 for MCT1 inhibition (Figure 2A). The treatment of either SR13800 or AZD3965 knowingly decreased the viability of the BM1-shBACH1 and MB436-shBACH1 cells in colony formation assays (Figure 2B,C and Appendix A) and viability assays using Calcein AM when compared to controls (Figure 2D,E). When BACH1 expression was rescued by the re-expression of BACH1 in shBACH1 cells, the viability of the shBACH1 cells was re-established (Figure 2D–F and Appendix A). These data indicate that MCT1 inhibition effectively suppresses the viability of shBACH1 compared to controls, and that BACH1 expression is sufficient to abolish the effects of SR13800 and AZD3965 on TNBC cell viability.

We next sought to determine whether the downstream of lactate catabolic pathways contributes to the vulnerability of shBACH1 cells by inhibiting LDHB or MPCs with small molecules, GSK2837808A and UK5099, respectively [28,40]. The treatment with either agent markedly reduced the colony formation of BM1-shBACH1 cells versus controls (Appendix A). Likewise, the inhibition of MPCs using UK5099 better suppressed the viability of MB436-shBACH1, MB231-shBACH1, and BM1-shBACH1 cells compared to controls (Appendix A). The transient silencing of *LDHB* mRNA using siRNA transduction consistently reduced the viability of BM1-shBACH1 cells versus controls (Appendix A). These data support the concept that inhibiting either step of the lactate catabolic pathways effectively suppresses the viability of BACH1-depleted TNBC cells.

### 3.3. Hemin Increases Efficacy of MCT1 Inhibitors through BACH1 Degradation

Intracellular heme levels are balanced by heme oxygenase 1 (HMOX1 or HO1) that breaks down heme to biliverdin and carbon monoxide (CO), and HMOX1 is transcriptionally suppressed by BACH1. Excessive heme binds to BACH1 protein for nuclear export and ubiquitin dependent degradation, subsequently increasing HMOX1 expression to further reduce heme [41]. Previously, hemin, also known as exogenous free heme, effectively induces the degradation of BACH1 protein in breast tumors in vivo, as well as breast and lung cancer cells, to subsequently modify cancer metabolism in a BACH1-dependent manner [32,33,34]. Therefore, we tested whether hemin treatment regulates lactate catabolic pathways through BACH1 degradation in TNBC cells. Similar to the genetic knock-down of BACH1 using shRNA, hemin treatment also resulted in the degradation of BACH1 proteins and increased MCT1 protein levels in BM1 and MB436 cells (Figure 3A and Appendix A). As MCT1 and LDHB proteins are induced by BACH1 degradation through hemin, hemin-treated BM1 cells that were cultured in the lactate-supplemented media (1.25 mM glucose, 4 mM glutamine, 4 mM lactate) also displayed a higher basal OCR with a lower ECAR than vehicle-treated BM1 cells (Figure 3B). Additionally, lactate infusion increased the OCR of BM1 and MB436 cells that were pre-treated with hemin for 48 h, compared to vehicle-treated cells (Figure 3C).

To exclude possible off-target effects of hemin, we adapted mutant Bach1 (mut Bach1) cells that express murine Bach1 with point mutations on heme binding sites that do not degrade Bach1 with hemin [34]. Mut Bach1-expressing BM1 or MB436 cells did not respond to hemin treatment in terms of Bach1 degradation, cell proliferation, metabolic gene regulation, or drug responses in our previous study [34]. In the presence of hemin, mut Bach1 cells did not alter OCR after lactate infusion, since mut-Bach1 cells cannot regulate BACH1, MCT1, or LDHB expression (Figure 3D and Appendix A). We then monitored the LDHB activity of TNBC cells that were pre-treated with hemin overnight. The BM1 and MB436 cells significantly increased LDHB activity upon hemin treatment, unlike MB436 mut-Bach1 cells (Figure 3E). These data indicate that hemin treatment modifies mitochondrial metabolism utilizing lactate through BACH1 degradation, suggesting that hemin could serve to generate the metabolic vulnerability of TNBC cells.

Next, we tested the effects of hemin when combined with drugs that inhibit the lactate transporter MCT1 on cell viability. Of note, the exposure of cancer cells to hemin (20 μM) does not affect proliferation or viability, as shown previously [34]. In colony formation assays, hemin plus SR13800, or hemin plus AZD3965, significantly reduced colony formation in wild type (wt)-Bach1 expressing MB436 cells when compared to each agent alone (Figure 3F). In contrast, hemin plus MCT1 inhibitors did not affect colony formation of mut-Bach1 cells when compared to the single treatment of MCT1 inhibitors (Figure 3G). These results indicate that hemin treatment could increase the efficacy of MCT1 inhibitors in suppressing viable cancer cells. Using mut Bach1 and wt Bach1 cells (controls), we carried out viability assays using hemin plus MCT1 inhibitors. The MCT1 inhibitors with hemin co-treatment significantly decreased the viability of wt Bach1 cells when compared to a single treatment with either agent, and the combination did not further decrease the viability of mut Bach1 cells when compared to treatment with only SR13800 or AZD3965 (Figure 4A and Appendix A). Furthermore, hemin plus AZD3965 showed a higher efficacy to suppress cell viability using multiple TNBC cell lines such as MB436, MB231, HCC1937, and 4T1.2 (Appendix A).

Due to the possibility of an inactive lactate catabolic pathway when lactate is depleted, we next tested the efficacy of the co-treatment of hemin and MCT1 inhibitors for cells cultured in lactate-free high glucose culture media (25 mM glucose, 2 mM glutamine DMEM). The combination treatment of hemin and MCT1 inhibitors without lactate supplementation did not elicit a higher efficacy for cell viability compared to either SR13800 or AZD3965 single treatment (Appendix A).

The blockade of LDHB and MPCs using small inhibitory molecules GSK2837808A and UK5099, respectively, better suppressed the viable cells when the molecules were co-treated with hemin (Appendix A). In parallel, treatment with UK5099, an MPC inhibitor, suppressed the colony formation of BM1 and MB436 with hemin combination (Appendix A). Similarly, lactate deprivation from the culture media abolished the combinatorial efficacy of GSK2837808A or UK5099 when compared with the single-agent treatment (Appendix A). Taken together, hemin treatment enhanced the efficacy of lactate metabolism-targeting drugs through BACH1 degradation in TNBC cells. These data suggest that the combination of MCT1 inhibitors with hemin might enhance the efficacy of either SR13800 or AZD3965 as cancer therapeutics.

### 3.4. Combining Hemin with MCT1 Inhibitors Effectively Suppresses TNBC Growth

We next explored whether BACH1-depleted tumors are more susceptible to MCT1 inhibition in vivo. Using a xenograft mammary tumor model that was orthotopically injected with control or BM1-shBACH1 cells, we monitored primary tumor growth after AZD3965 treatment. When tumors grew to a palpable size, female athymic nude mice (*n* = 3–6/group) received AZD3965 (30 mg/kg body weight) by oral gavage daily, and tumor growth was measured weekly using a caliper. The AZD3965-treated mice showed decreased volumes of BM1-shBACH1 tumors, but not control (shCont) tumors (Figure 4B and Appendix A). Since AZD3965 decreases tumor metastasis in preclinical melanoma models [9], in this mouse model, we also quantified cancer cells in circulation. Intravasation assays showed that AZD3965 treatment inhibited the circulating shBACH1 tumor cells in mice compared to those treated with vehicle control, or shCont tumors (Appendix A). Similarly, the hematoxylin and eosin (H&E) staining of lung tissues isolated from these mice demonstrated that AZD3965 treatment significantly reduced the micro-metastases of shBACH1 tumors to the lungs (Appendix A).

As hemin is a potent BACH1 inhibitor, we then tested whether pharmacologic inhibition of BACH1 using hemin could elicit tumor suppression in vivo similar to shBACH1 when co-treated with an MCT1 inhibitor. We used a xenografted mammary tumor model using female athymic nude mice (*n* = 8–9/group) that were orthotopically injected with BM1 cells. Hemin administration (50 mg/kg body weight) was initiated 10 days in advance to degrade tumoral BACH1 proteins, and then this administration was combined with SR13800 (30 mg/kg body weight) daily, both intraperitoneally injected. The combination treatment of hemin and SR13800 suppressed primary tumor volumes when compared to a single treatment of SR13800 (Figure 4C and Appendix A). In addition, the effect of SR13800 was validated by the detection of reduced MCT1 expression in the tumor lysates using Western blotting (Appendix A). Notably, another study validated that the long-term (30 days) treatment of SR13800 effectively reduced breast tumors [19]. Among all animal experiments, we did not observe any noticeable side effects of the drugs, including body weight loss or behavioral changes (Appendix A), as previously investigated [19]. Therefore, our results suggest that BACH1 inhibition helps increase the efficacy of MCT1 inhibitors for tumor suppression in vivo, recommending a novel combination therapy using hemin and MCT1 inhibitors.

Next, we analyzed breast tumor data from the Cancer Genome Atlas (TCGA) patient cohorts (*n* = 980) for BACH1, MCT1, and LDHB mRNA expression. A heat map dictates upregulated (red) or downregulated (blue) expression levels of MCT1 (SLC16A1) and BACH1 mRNA, indicating a potential inverse correlation between BACH1 and MCT1 expression (Appendix A). The downstream genes of lactate catabolic pathways, LDHA and LDHB, also showed potential inverse correlations with BACH1 expression in breast tumors. These data support our findings showing that BACH1 suppresses lactate catabolic pathways through MCT1 and LDHB inhibition in TNBC cells.

## 4. Discussion

Our work showed that BACH1 depletion increases the lactate-dependent mitochondrial respiration of TNBC cells and cell death upon lactate transporter inhibition. Notably, BACH1 depletion does not affect cell survival or proliferation [34,42]. We demonstrated that BACH1 is a newly identified transcriptional regulator of lactate catabolic pathways, as well as an endogenous determinant of TNBC response to MCT1 inhibitors. Therefore, our study suggests that targeting BACH1 projects a novel approach for enhancing the efficacy of inhibitors targeting lactate catabolic pathways. Our findings suggest that the co-inhibition of BACH1 and MCT1 by re-purposing an FDA-approved drug, hemin, has therapeutic potential for TNBC tumors.

The lactate transporter MCT1-dependent lactate provides carbons and energy for lung adenocarcinoma proliferation [5,15]. Lactate is a more favored major substrate over glycolysis-driven pyruvate for the mitochondrial metabolism of cancer cells [43]. In addition, lactate becomes the carbon source for breast cancer cells when glucose is limited [14]. Our current study identified that BACH1 suppresses lactate catabolism through inhibiting the expression of the lactate transporters MCT1 and LDHB, and further suppresses mitochondrial respiration using lactate in TNBC cells. Our previous study demonstrated that BACH1 is sufficient to secrete lactate after aerobic glycolysis [34]. These results reveal that BACH1 is an endogenous key player in determining whether cancer cells produce or consume lactate based on the nutrient availability.

The lactate metabolic proteins MCT1 or LDHB are effective targets for suppressing tumorigenesis [28]. We found that targeting lactate catabolic pathways, including MCT1, is more effective for tumor suppression when BACH1 levels are low. Although MPCs are not directly regulated by BACH1, the blockade of MPCs using a small inhibitor with hemin more effectively reduced cancer cell viability. This is possibly due to the forced lactate catabolic flows using MCT1 and from LDHB activity due to BACH1 loss. These findings suggest that the activation of lactate utilization by cancer cells is a key strategy for enhancing the efficacy of lactate catabolism-targeting drugs as a cancer therapy. We could create synthetic lethality by adding hemin with a lower dosage of MCT1 and LDHB inhibitors, although MCT1 and LDHB inhibitors at higher concentrations are reported to suppress cancer growth [18,19,27,29]. Importantly, the deprivation of the available lactate in the growth media abolished the efficacy of the inhibitors targeting MCT1, LDHB, or MPCs, even in the presence of hemin. We surmise that the lactate-abundant tumor microenvironment in vivo helps to activate the lactate catabolic pathways of cancer cells, thus promoting the efficacy of lactate catabolism-targeting inhibitors.

MCT1 expression is inversely associated with improved patient outcomes, highlighting its potential use as a biomarker for anti-cancer therapy [18,20]. Both small molecules (SR13800 and AZD3965) that the target MCT1 showed anti-tumor effects on breast tumor growth in pre-clinical models [18,19,28], and AZD3965 is now being tested in clinical trials. At a lower dose of AZD3965 (100 mg/kg vs. 30 mg/kg), notably, we did not observe tumor suppression in our xenograft model. At a similar dose, but with a shorter-term treatment of SR13800 (11 vs. 30 days), we did not observe any tumor-suppression effects by using a single SR13800 in our study. However, adding a BACH1 drug, hemin, produced higher efficacy for tumor suppression compared to the single treatment of both MCT1 inhibitors, both at a lower dosage or in a shorter term. These data suggest that the combination of hemin and MCT1 inhibitors could be a potential strategy to reduce the adverse effects of MCT1 inhibitors by reducing their dose and time.

Previously, other groups reported that MCT1 is critical for the pyruvate export of glycolytic breast cancer cells [18,44]. The blockade of MCT1 using AZD3965 decreased breast cancer cells proliferation by suppressing pyruvate export, but increased mitochondrial oxygen consumption. These data raise the question of whether MCT1 expression induced by BACH1 depletion mediates pyruvate export from TNBC cells. In our study, however, BACH1 depletion increased mitochondrial respiration using extracellular lactate, but not using extracellular pyruvate, as measured by OCR. Additionally, BACH1 depletion increased the levels of pyruvate from glucose through ^13^C_6_-labeled glucose tracing analysis in our previous study [34]. In the BACH1-depleted TNBC cell lines that we tested, glucose-driven pyruvate was mostly utilized by pyruvate dehydrogenase (PDH) and its kinase (PDK), which are directly regulated by BACH1 for mitochondrial respiration. Moreover, pyruvate addition rescued mitochondrial respiration by balancing the NAD^+^/NADH ratio in BACH1-depleted TNBC cells, ameliorating the suppressive effects of a mitochondrial inhibitor, metformin [34,45]. Based on our data, MCT1 induction by BACH1 depletion is likely not involved in pyruvate export in TNBC, as BACH1 plays a major role in mitochondrial biogenesis in TNBC cells. However, AZD3965 might regulate the pyruvate transport of cancer cells, which could be further analyzed in the future. As lactate is the main component that is shared by many immune cells within the breast tumor microenvironment, including fibroblasts and cancer cells [13,46,47], and since MCTs transport lactate and pyruvate bidirectionally [22], examining the dynamics of lactate release and uptake between cancer cells and immune cells in a tumor microenvironment could be the focus of future investigations.

## Figures and Tables

**Figure 1 cells-11-01177-f001:**
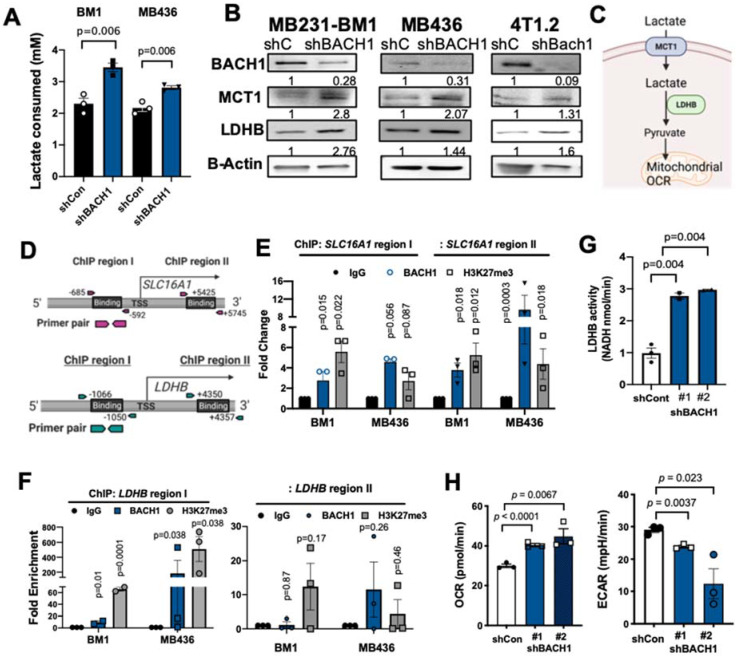
BACH1 suppresses MCT1 transcription activity and lactate catabolism. (**A**) Lactate consumption by BM1-shBACH1 and control cells. (**B**) Representative protein blots of BACH1, MCT1, LDHB, and beta-Actin in BM1-shBACH1, MDA-MB-436-shBACH1, and 4T1.2-shBach1 cells relative to their control (shCont) cells. Band density is indicated above the band in Western blots. (**C**) Diagram showing lactate catabolic pathway. (**D**) Schematic diagram of BACH1 binding regions on the promoter of SLC16A1 and LDHB for ChIP assays. (**E**) Relative fold changes of BACH1, H3K27me3, and IgG on the SLC16A1 promoter regions using BM1 and MB436 cells. (**F**) Relative fold changes of BACH1, H3K27me3, and IgG on the LDHB promoter regions using BM1 and MB436 cells. (**G**) Relative LDHB activity in control and BM1-shBACH1 cells (#1, #2 shRNA clones). (**H**) Measurement of OCR (left) and ECAR (right) in control and BM1-shBACH1 cells. Mean + s.e.m., *n* = 3 biologically independent samples. Two-tailed *t*-test.

**Figure 2 cells-11-01177-f002:**
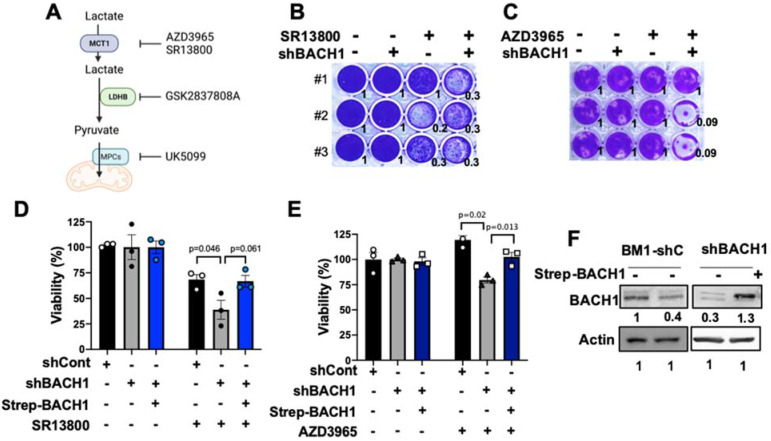
BACH1-depleted TNBC cells are more sensitive to the MCT1 inhibitors for cell viability than control cells. (**A**) Diagram showing lactate catabolic pathways with small inhibitory molecules. AZD3965 and SR13800 inhibit MCT1, GSK2837808A inhibits LDHB, and UK5099 inhibits MPCs. (**B**,**C**) Colony formation of BM1-shBACH1 and control cells treated with SR13800 (50 μM) or AZD3965 (100 μM). After 48 h of drug treatment, cells were further incubated in DMEM (25 mM glucose, 2 mM glutamine) media for 10 days and stained using crystal violet. Quantified values of stained cells in the three replicates are shown. (**D**,**E**) Cell viability (%) of BM1-shBACH1 or re-expressed BACH1 in shBACH1 cells relative to the shControl cells that were treated with SR13800 (50 μM) or AZD3965 (100 μM) for 48 h and Calcein AM staining. Mean (*n* = 3/cells) ± s.e.m., *p*-values by two-tailed *t*-test. (**F**) Representative protein blots of BACH1 and beta-Actin as a loading control of cells used for (**D**,**E**).

**Figure 3 cells-11-01177-f003:**
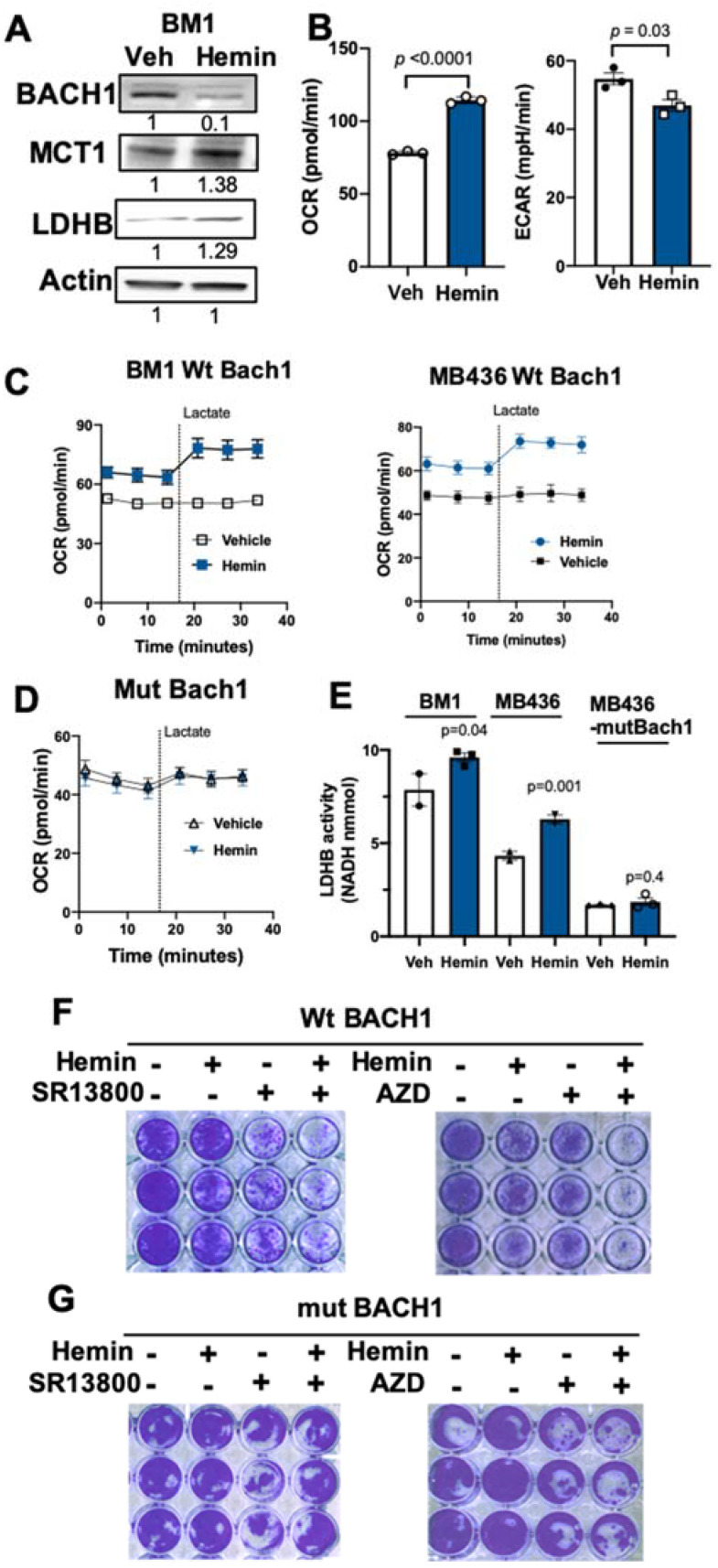
Hemin increases efficacy of MCT1 inhibitors through BACH1 degradation. (**A**) Representative protein blots of BACH1, MCT1, and beta-Actin in BM1 cells treated with hemin (20 μM) or vehicle for 48 h. (**B**) Measurement of basal OCR and ECAR of BM1 cells pre-treated with hemin overnight. (**C**,**D**) Measurement of OCR of BM1-Wt BACH1, MB436-Wt BACH1, or BM1-mut Bach1 cells infused with lactate (4 mM). For (**B**–**D**), mean (*n* = 3/cells) ± s.e.m., *p*-values by two-tailed *t*-test. (**E**) Relative LDHB activity of BM1, MB436, and MB436-mut Bach1 cells that were treated with hemin (20 μM) for 48 h (left). Representative protein blots showing BACH1 expression levels in cell lines (right). Mean (*n* = 2–3/cells) ± s.e.m., *p*-values by two-tailed *t*-test. (**F**,**G**) Colony formation of BM1-wt BACH1 or mut Bach1 cells that are treated with either hemin, SR13800 (50 μM), AZD3965 (100 μM), hemin+SR13800, or hemin+AZD3965 for 48–72 h.

**Figure 4 cells-11-01177-f004:**
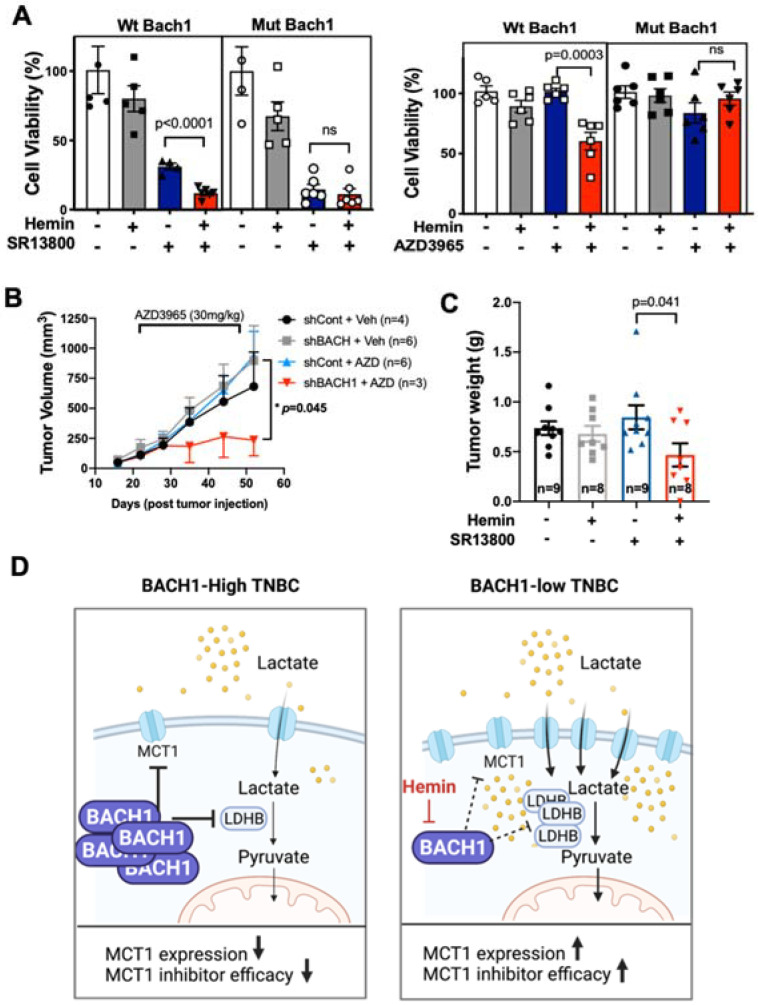
Combination treatment of hemin and the MCT1 inhibitors is effective to suppress TNBC growth. (**A**) Viability (%) of BM1-wt BACH1 or mut Bach1-expressing cells treated with hemin, SR13800 (50 μM), hemin+SR13800, or AZD3965 (100 μM), hemin+AZD3965. After incubation of cells for 48 h, viable cells were stained using Calcein AM; mean (*n* = 6/cells) ± s.e.m., *p*-values by two-tailed *t*-test. (**B**) Tumor volumes from athymic nude mice orthotopically injected with control and BM1-shBACH1 cells (shCont *n* = 4, shBACH1 *n* = 6, shCont+ AZD3965 *n* = 6, shBACH1+AZD3965 *n* = 3) and treated with AZD3965 (100 mg/kg body weight) by oral gavage for 17 days. (**C**) Tumor weights from athymic nude mice orthotopically injected with BM1 cells and treated with hemin (50 mg/kg body weight) daily by intraperitoneal injection and/or SR13800 for 10 days (vehicle *n* = 9, hemin *n* = 8, SR13800 *n* = 9, hemin + SR13800 *n* = 8). Representative tumor images are shown (below). (**D**) Schematic diagram indicating lactate catabolic pathways regulated by BACH1 in TNBC cells.

## Data Availability

All data are available in the main text or the Appendix A upon request.

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
