# Peer review of "A Heme-Binding Transcription Factor BACH1 Regulates Lactate Catabolism Suggesting a Combined Therapy for Triple-Negative Breast Cancer"

_cells, 2022, doi:10.3390/cells11071177_

Round 1
Reviewer 1 Report
In the manuscript entitled “A heme-binding transcription factor BACH1 regulates lactate catabolism in triple-negative breast cancer”, the authors identified a new therapeutic avenue in TNBC, based on the depletion of BACH1 using either shRNA or a non-toxic inhibitor hemin in order to increase efficacy of MCT1 inhibitors, and further inhibit lactose metabolism. The manuscript theme is of interest to the oncology field, the experimental approach is appropriate to test the authors’ hypothesis and the conclusions are drawn from the results obtained.
However, a few aspects should still be checked.
Abstract and Introduction
Page 1, line 41: please, include examples of metabolic targeting drugs to which cancer cells are resistant to.
Page 3, section Cell Lines: please, specify every cell line used in this work.
Page 6, line 294: please, replace BAHC1 with BACH1.
All in all, these sections are clear and well written.
Methods
This section is clear and well written.
Results
Figure 1 B. WB quantification is missing.
Figure 2 B&C. Colonies quantification is missing. Please, include. It is unclear what #1, #2 and #3 mean in the colony formation assays figures. Please, include it in the legend.
Figure 2 F. WB quantification is missing.
Supplementary figure 5 H. Please, include a correlation analysis using the TCGA dataset between every gene of interest, to reinforce what is observed in the heatmap.
Figure 3 A. WB quantification is missing.
It has been shown in androgen independent prostate cancer cells that hemin decreases not only the ECAR (as seen in this work), but also the OCR (as opposed to what the authors report in this manuscript). Since prostate and breast cancers that progress are hormone independent, and in both cases hemin has been reported as an antitumoral drug, the authors should discuss these findings in the Discussion section.
Author Response
Referee 1: In the manuscript entitled “A heme-binding transcription factor BACH1 regulates lactate catabolism in triple-negative breast cancer”, the authors identified a new therapeutic avenue in TNBC, based on the depletion of BACH1 using either shRNA or a non-toxic inhibitor hemin in order to increase efficacy of MCT1 inhibitors, and further inhibit lactose metabolism. The manuscript theme is of interest to the oncology field, the experimental approach is appropriate to test the authors’ hypothesis and the conclusions are drawn from the results obtained.
However, a few aspects should still be checked.
Abstract and Introduction
Page 1, line 41: please, include examples of metabolic targeting drugs to which cancer cells are resistant to.
We thank the reviewer for the comment, and have added metabolic pathways that are targeted using small inhibitory molecules such as serine biosynthesis and guanosine biosynthesis pathways of cancer cells which provides therapeutic vulnerabilities.
Page 3, section Cell Lines: please, specify every cell line used in this work.
We thank the reviewer for the comment, and specified every cell lines in materials and method as suggested.
Page 6, line 294: please, replace BAHC1 with BACH1.
We thank the reviewer for this comment and have now corrected typo.
All in all, these sections are clear and well written.
We appreciate this reviewer’s comment and support.
Methods
This section is clear and well written.
We appreciate this reviewer’s comment and support.
Results
Figure 1 B. WB quantification is missing.
We have now added quantification below the blots in Figure 1B.
Figure 2 B&C. Colonies quantification is missing. Please, include. It is unclear what #1, #2 and #3 mean in the colony formation assays figures. Please, include it in the legend.
We have now added quantification below the images and #1,2,3 replicates were indicated in the Figure legends and Materials & Methods.
Figure 2 F. WB quantification is missing.
We have now added quantification below the images and indicated in the Figure legends.
Supplementary figure 5 H. Please, include a correlation analysis using the TCGA dataset between every gene of interest, to reinforce what is observed in the heatmap.
We thank the reviewer for the comment but at present the heatmap indicates up-regulated or down-regulated expression of genes in the TCGA without correlation analyses. The heatmap is to show inverse expression patterns of genes in breast tumors.
Figure 3 A. WB quantification is missing.
We have now added quantification below the images and indicated in the Figure legends.
It has been shown in androgen independent prostate cancer cells that hemin decreases not only the ECAR (as seen in this work), but also the OCR (as opposed to what the authors report in this manuscript). Since prostate and breast cancers that progress are hormone independent, and in both cases hemin has been reported as an antitumoral drug, the authors should discuss these findings in the Discussion section.
We thank the reviewer for the comment and as our study at present is focused on breast cancer, no other cancer types that alter metabolism by hemin treatment are mentioned, although its relevance and broad application of hemin in other cancer types. However, we will further review application of hemin to target cancer metabolism including androgen independent prostate cancer as reviewer mentioned.
Reviewer 2 Report
This article were described the novel suggestion of hemin on BACH1-mediated regulation of lactate catabolism in TNBC. To elucidate BACH1 is new candidates for TNBC treatment drugs, authors were examined signaling pathway of lactate-MCT1-hemin-BACH1 and -mitochondria with hemin, specific-antagonists, targeted shRNAs and animal experiment.
-Line 370 etc., a BACH1 drug, this is not clear words. please clearfy.
- BM1 cells, no information was provided in materials and methods. Please check.
- almost data were designed for clearfy the mechanism of lactate-MCT1-BACH1 pathway with specific antagonist to key molecules, whereas not compared parallelly hemin effects in each panel. In addition, authors were strongly suggested the hemin effects as new drug candidate against TNBC but not strong actually. These points can solve by changing the FOCUS of title and abstracts to combined therapy with previously guaranteed drugs, I suggest.
Author Response
Referee 2: This article were described the novel suggestion of hemin on BACH1-mediated regulation of lactate catabolism in TNBC. To elucidate BACH1 is new candidates for TNBC treatment drugs, authors were examined signaling pathway of lactate-MCT1-hemin-BACH1 and -mitochondria with hemin, specific-antagonists, targeted shRNAs and animal experiment.
-Line 370 etc., a BACH1 drug, this is not clear words. please clearfy.
We clarified this line 370 by replacing a BACH1 drug to hemin.
- BM1 cells, no information was provided in materials and methods. Please check.
We defined BM1 cells in the main manuscript and materials.
- almost data were designed for clearfy the mechanism of lactate-MCT1-BACH1 pathway with specific antagonist to key molecules, whereas not compared parallelly hemin effects in each panel. In addition, authors were strongly suggested the hemin effects as new drug candidate against TNBC but not strong actually. These points can solve by changing the FOCUS of title and abstracts to combined therapy with previously guaranteed drugs, I suggest.
We thank the reviewer for insightful comments and agree that the focus of our study is the mechanism of lactate and BACH1, MCT with specific antagonist. To this end, we modified abstract and the title adding a combined therapy.